# The Supplementation of Berberine in High-Carbohydrate Diets Improves Glucose Metabolism of Tilapia (*Oreochromis niloticus*) via Transcriptome, Bile Acid Synthesis Gene Expression and Intestinal Flora

**DOI:** 10.3390/ani14081239

**Published:** 2024-04-20

**Authors:** Hongyu Liu, Menglin Wei, Beiping Tan, Xiaohui Dong, Shiwei Xie

**Affiliations:** 1Laboratory of Aquatic Animal Nutrition and Feed, College of Fisheries, Guangdong Ocean University, Zhanjiang 524088, China; liuhyu@gdou.edu.cn (H.L.); weiml7@163.com (M.W.); dongxiaohui2003@163.com (X.D.); xiesw@gdou.edu.cn (S.X.); 2Aquatic Animals Precision Nutrition and High-Efficiency Feed Engineering Research Center of Guangdong Province, Zhanjiang 524088, China; 3Key Laboratory of Aquatic, Livestock and Poultry Feed Science and Technology in South China, Ministry of Agriculture, Zhanjiang 524088, China

**Keywords:** *Oreochromis niloticus*, berberine, bile acid, intestinal flora, glucose metabolism

## Abstract

**Simple Summary:**

An excessive carbohydrate content in diets can have a negative effect on fish growth. In this study, the effects of berberine supplementation in high-carbohydrate diets on the growth performance, glucose metabolism, bile acid synthesis, liver transcriptome, and intestinal flora of Nile tilapia were investigated. Conclusion: An appropriate amount of berberine can stimulate the synthesis of bile acids by changing the structure of the intestinal flora of tilapia, thus promoting glycolysis and inhibiting gluconeogenesis, then regulating the blood glucose, which is beneficial to the growth of tilapia. This provides the data to improve carbohydrate utilization in tilapia and support the development of high-carbohydrate diets in aquaculture.

**Abstract:**

Berberine is an alkaloid used to treat diabetes. This experiment aimed to investigate the effects of berberine supplementation in high-carbohydrate diets on the growth performance, glucose metabolism, bile acid synthesis, liver transcriptome, and intestinal flora of Nile tilapia. The six dietary groups were the C group with 29% carbohydrate, the H group with 44% carbohydrate, and the HB1-HB4 groups supplemented with 25, 50, 75, and 100 mg/kg of berberine in group H. The results of the 8-week trial showed that compared to group C, the abundance of Bacteroidetes was increased in group HB2 (*p* < 0.05). The cholesterol-7α-hydroxylase (CYP7A1) and sterol-27-hydroxylase (CYP27A1) activities were decreased and the expression of FXR was increased in group HB4 (*p* < 0.05). The pyruvate carboxylase (PC) and phosphoenolpyruvate carboxykinase (PEPCK) activities was decreased in group HB4 (*p* < 0.05). The liver transcriptome suggests that berberine affects carbohydrate metabolic pathways and primary bile acid synthesis pathways. In summary, berberine affects the glucose metabolism in tilapia by altering the intestinal flora structure, enriching differentially expressed genes (DEGs) in the bile acid pathway to stimulate bile acid production so that it promotes glycolysis and inhibits gluconeogenesis. Therefore, 100 mg/kg of berberine supplementation in high-carbohydrate diets is beneficial to tilapia.

## 1. Introduction

With the increase in population, aquaculture plays a vital role in ensuring food security in China [1]. Freshwater fish farming production reached 27.1 million tons in 2022, an increase of 2.66% compared to 2021 in China. However, the price of fishmeal and other feed ingredients is rising. In order to reduce the cost of aquaculture and improve its economic performance, some scholars believe that the use of carbohydrates is a good choice. Fish have a “diabetic” constitution, and an excessive carbohydrate content in the diet may cause problems such as liver damage and a reduced immune performance in fish, affecting growth performance [2]. Therefore, determining how to improve the utilization of carbohydrates in fish has become a hot research topic in aquaculture. Some additives such as Chinese herbs can improve the tolerance of fish to carbohydrates and are widely used in diets because they are natural, have low toxicity and low drug resistance [3]. Berberine is an isoquinoline alkaloid extracted from the traditional Chinese herb *Coptis chinensis* Franch, with a bitter taste [4]. Berberine has an important role in antibacterial, antipyretic, and antilipidemic treatments, and the treatment of diabetes [5]. By regulating the bile acid balance, berberine reduces insulin resistance; the mechanism of action may be due to a decrease in the number of intestinal flora capable of producing bile salt hydrolase [6]. Houjun Pan et al. added 30 mg/kg of berberine to the diet of grass carp and showed that berberine may affect serum glucose through the structural regulation of the gut microbiota [7].

The intestinal flora is influenced by diet and the environment. It has been suggested that the intestinal flora is an important “organ” of the organism [8] that not only plays an important role in immunity, but also regulates plastic and energy metabolism, and promotes the digestion and absorption of nutrients [9]. Diabetes, metabolic disorders, and the disruption of the gut microecological balance can all result from high-carbohydrate diets [10]. The intestinal flora influences the host by producing metabolites such as short-chain fatty acids, tryptophan metabolites and bile acids. The vast majority of intestinal bacteria such as Bacteroidetes and Lactobacillus produce bile salt hydrolase (BSH), an important product that affects bile acid uncoupling, and these highly active flora can influence bile acid metabolism [11]. Bile acids, in turn, can react to the intestinal flora; for example, the diversity of the flora is affected by primary bile acids, and secondary bile acids affect the ratio of Firmicutes to Bacteroidetes [12]. Bile acids are metabolites of cholesterol and are divided into primary and secondary bile acids. The liver is the only place where primary bile is synthesized, and secondary bile acids are products of the intestinal flora [13]. Primary bile acid synthesis is divided into classical and alternative pathways. The production of cholestatic deoxycholic acid (CDCA) and cholic acid (CA) from cholesterol is catalyzed by cholesterol-7α-hydroxylase (CYP7A1) in the classic pathway. The other pathway involves the catalysis of sterol-27-hydroxylase (CYP27A1) from cholesterol to 27-hydroxycholesterol, which is then further processed by oxysterol-7α-hydroxylase (CYP7B1) to produce CDCA. The ratio of bile acids produced by the two pathways is determined by sterol-12α-hydroxylase (CYP8B1) [14]. Bile acids are stored in the gallbladder before eating and are secreted into the intestine during eating to be metabolized by the intestinal flora into secondary bile acids [15]. The liver–gut axis is a continuous circulation process through which bile acids are produced in the liver, enter the intestine, and then are reabsorbed at the end of the ileum and returned to the liver [16].

The Nile tilapia, a tropical fish from Africa, is one of the main species of freshwater aquaculture in southern China. It grows fast, has strong environmental adaptability, good meat quality, and is widely popular with businesses and consumers [17]. A diet containing 30~35% carbohydrates is better for tilapia growth [18]. However, fewer studies have been conducted on the glucose metabolism of tilapia by exploring the effects of berberine on the intestinal flora and bile acids. This experiment explored the effects of berberine supplementation in high-carbohydrate diets on the glucose metabolism in tilapia by liver transcriptome, intestinal flora and bile acid synthesis.

## 2. Materials and Methods

### 2.1. Experimental Diets and Animals

In this experiment, diets were made with whitefish meal and casein as protein sources, fish oil and soybean phospholipid oil as lipid sources, as well as starch and corn starch as carbohydrate sources, and the formulation and proximate analysis of the trial diets are shown in Table 1. The diets were categorized into normal-carbohydrate-content diets (C) and high-carbohydrate-content diets (H), and different contents (25, 50, 75, and 100 mg/kg) of berberine (C_20_H_18_ClNO_4_, Maclean) were supplemented to the high-carbohydrate diets in the HB1-HB4 groups; the amount of berberine supplementation was set with reference to previous studies [19,20]. Firstly, the diet was crushed with a grinder and passed through a 60-mesh sieve. Then, it was added to a V-mixer and combined thoroughly. Next, it was combined once more with water in a food-grade mixer. Finally, the diet was made into 2.5 mm pieces by a twin-screw extruder. After 48 h of drying indoors, the diet was stored in a refrigerator at −20 °C. The diet composition was determined according to the AOAC (1995) standard [21].

One thousand tilapia were purchased from a tilapia fish farm in Guangdong. After three weeks of temporary feeding with a commercial diet in an outdoor cement pond at the Biological Research Base of Guangdong Ocean University, 540 tilapia, weighing an average of 4.74 g, were arranged in 18 × 1 m^3^ net boxes at random, with 30 fish in each box. These fish were divided into six groups, Group C (0 mg/kg), Group H (0 mg/kg), Group HB1 (25 mg/kg), Group HB2 (50 mg/kg), Group HB3 (75 mg/kg), and Group HB4 (100 mg/kg), with three copies in each group. They were fed twice a day to satiation at 8:00 and 17:00 for 8 weeks. The tilapia were cultivated in flow-through water with dissolved oxygen > 7 mg/L, pH 6.9–7.2, and a water temperature of 28–30 °C.

### 2.2. Sample Collection and Chemical Analysis

Tilapia were fasted for 24 h before samples were collected. During collection, the fish in each net box were counted and weighed in total, and then three fish were randomly selected and measured for body length, body weight, liver weight and viscera weight. The growth performance was calculated by the following formula [22]:Survival rate (SR, %) = final number of fish/initial number of fish × 100;
Weight gain rate (WGR, %) = (final mean weight − initial mean weight)/initial mean weight × 100;
Specific growth rate (SGR, %/d) = (ln final weight − ln initial weight)/day × 100;
Feed conversion ratio (FCR) = total feed weight/(final weight − initial weight);
Hepatosomatic ratio (HSI, %) = (liver weight/fish weight) × 100;
Viscerosomatic index (VSI, %) = (viscera weight/fish weight) × 100;
Condition factor (CF, %) = 100 × body weight/body length^3^;

To begin with, three fish were randomly taken from each net box for blood sampling from the tail vein, and this blood was allowed to stand in the refrigerator at 4 °C for 12 h. It was then centrifuged at 3000× *g* at 4 °C for 10 min to obtain the serum, which was stored in a refrigerator at −80 °C for the subsequent determination of the indexes. Finally, the remaining tilapia were dissected, and part of the liver was preserved in RNA-Later; furthermore, the other part of the liver and hindgut were rapidly frozen in liquid nitrogen, then transferred to a −80 °C refrigerator for sample analysis.

Referring to our previous research [23,24], the serum glucose (GLU), triglycerides (TG) and total cholesterol (T-CHO), superoxide dismutase (SOD), acid phosphatase (ACP) and alkaline phosphatase (AKP), hepatic and intestinal bile acid content, as well as hepatic glycogen and muscle glycogen content, were measured using commercial kits (Nanjing Jianjian Institute of Biological Engineering, Nanjing, China). Hepatic glucose metabolism enzyme activities such as hexokinase (HK), pyruvate kinase (PK), 6-phosphofructokinase (PFK), pyruvate carboxylase (PEPC), phosphoenolpyruvate carboxylase (PEPCK), glucose-6-phosphate phosphatase (G6Pase), and the rate-limiting enzymes for bile acid synthesis, such as cholesterol 7α hydroxylase (CYP7A1), sterol 12α hydroxylase (CYP8B1) and sterol-27-hydroxylase (CYP27A1), were measured with a commercial kit (ELAS). The glucose-metabolizing enzymes and bile acid enzymes were entrusted to Shanghai Enzyme Link Biotechnology Co. (Shanghai, China).

### 2.3. Intestine Microbiological Analysis

Intestinal flora genomic DNA was extracted from the hindgut of the tilapia, and then the V3-V4 region of 16S rDNA was amplified with specific primers. The primer sequence was 341F (CCTACGGGGNGGCWGCAG) 806R (GGACTACHVGGGGTATCTAAT). The amplification products were then recovered and quantified, sequencing libraries were constructed and sequenced online, and the flora data were analyzed on the omicsmart platform. The sequencing and data analysis of the intestinal flora were entrusted to Guangzhou Kidio Biotechnology Co. (Guangzhou, China).

### 2.4. Histological Analysis of Liver

Fresh tilapia liver was taken and fixed by submerging it in Bouin’s fixative solution, dehydrated with alcohol and embedded in paraffin, sectioned and finally stained with hematoxylin. After the liver HE sections were made, they were observed and photographed with an inverted microscope (Olympus, Tokyo, Japan), at 40×10×. The HE sections of the liver of the tilapia were entrusted to Wuhan Sevier Biotechnology Co. (Wuhan, China).

### 2.5. Real-Time Quantitative RT-PCR Analysis of Gene Expression

RNA from the liver was extracted using the TransZol Up Plus RNA (Beijing TransGen Biotech Co., Ltd., Beijing, China) kit, and the quality of the RNA was detected by 1.2% agarose gel electrophoresis; the extracted RNA was reverse-transcribed into cDNA and then stored in a −80 °C refrigerator for use in the subsequent RT-PCR experiments. Tilapia hexokinase (HK), hepatic-type phosphofructokinase (PFKl), glucose-6-phosphate catalytic subunit (G6PCA), phosphoenolpyruvate carboxylase (PCK), glycogen synthase (GYS), glucose transporter protein 2 (GLUT2), cholesterol 7-α-hydroxylase (CYP7A1), sterol 12-α-hydroxylase (CYP8B1), and farnesol X receptor (FXR) genes were selected for amplification. The primers were designed according to the results of the published tilapia sequences and transcriptomes, and are shown in Table 2. The qPCR experiments were performed using a 384-well plate with a LightCycler^®^ 480 II (Roche Diagnostics GmbH, San Francisco, CA, USA). Finally, the relative expression of the target genes was calculated by the 2^−ΔΔCt^ method.

### 2.6. Transcriptome Sequencing and Analysis 

Based on the WGR and FC, liver samples of tilapia from groups C, H, and HB4 (100 mg/kg) were selected for transcriptome analysis. The tilapia were dissected, and the livers were rapidly frozen in liquid nitrogen; then, RNA was extracted from the livers with TRIzol reagent, the quality of RNA was examined using agarose gel electrophoresis and a spectrophotometer, and finally the total RNA was enriched.

The library construction of the transcriptome was performed: mRNA was isolated and interrupted, cDNA duplexes were synthesized and purified, and end-repaired cDNAs were added with A-tails and sequencing connectors, and screened, amplified and purified. Differential genes were tested using *p* value (P) and foldchange (FC), and differences were screened for *p* < 0.05 and FC > 1.2. Genes were quantified and differentially analyzed, and genes with significant differences were used for GO and KEGG enrichment analysis. The transcriptome assay was entrusted to Guangzhou Kidio Biotechnology Co.

### 2.7. Challenge Test 

For the challenge test, ten tilapia were chosen at random from each group. Over the course of the five-day experiment, 200 μL of *Streptococcus agalactiae* at a concentration of 1 × 10^7^ cfn/mL was injected into each tilapia at the base of its pectoral fins; the bacteria were obtained from the Laboratory of Aquatic Animal Diseases of Guangdong Ocean University. At the end of the test, fresh blood was taken from the tilapia by tail vein phlebotomy and stored in a −80 °C refrigerator for the subsequent evaluation of the serum immunity indexes.

### 2.8. Statistical Analysis

All the results of the above experiments were statistically analyzed using IBM SPSS Statistics 22. One-way ANOVA was used to analyze the significance, and Dunnet’s test was used to determine whether the difference between the other groups was significant compared with group C. *p* < 0.05 indicated significance, and the results were expressed as mean ± standard error (Mean ± SE).

## 3. Results

### 3.1. Growth Performance

As can be seen from Table 3, there was no significant difference in the SR between the groups, which were all above 90%. Compared with group C, the WGR and SGR decreased in group H and improved with the increase in berberine supplementation, with no significant difference between groups. Compared with group C, the FCR increased in group H and decreased with berberine; however, there was no significant difference in the FCR between groups. The HSI of group H increased, compared with that in group C, and the VSI increased at a berberine supplementation of 25 mg/kg and decreased after a further increase in berberine supplementation; meanwhile, the HB4 group had the lowest HSI. There was no significant difference among the groups for the VSI and CF. The above results indicate that berberine supplementation in high-carbohydrate diets increases the WGR and SGR, and decreases the FC and HSI of tilapia.

### 3.2. Intestinal Flora Analysis

High-throughput sequencing of the hindgut flora of tilapia was performed with the following results. The analysis of the indicator species is shown in Figure 1, where the analysis of the VENN diagram shows that there are 192 core OTUs in the six groups, and that the ratio of core OTUs to group-specific OTUs was 23.67%, 35.82%, 44.65%, 48.73%, 47.64%, and 50.79%, respectively. The hindgut flora of the tilapia decreased with high dietary carbohydrate inclusion and further with increasing berberine supplementation in high-carbohydrate diets.

#### 3.2.1. Richness and Diversity Analysis

The α-diversity indices are shown in Table 4, and by analyzing the abundance of bacterial flora, it can be seen that Chao1 and ace were significantly lower in group H than in group C. Compared with group C, the sobs, Chao1 and ace indices were lower in all other groups and the differences were significant (*p* < 0.05). The Shannon and Simpson indices were not significantly different (*p* > 0.05) among the six groups. The high-carbohydrate diet significantly decreased the abundance of hindgut flora in tilapia, but the increase in berberine content had no significant effect on the abundance of flora.

The β-diversity is shown in Figure 2, and by analyzing the flora structure, group C can be observed to be spatially separated from the other groupings, and the Kru–Wall test shows that the difference is significant (*p* = 0.03). Although there is some distance in the space between the H-HB4 groups, the difference is not significant (*p* = 0.10). Compared with the C group, the high-carbohydrate diet significantly changed the spatial structure of the hindgut flora of tilapia, but berberine supplementation had no significant effect on the flora structure.

#### 3.2.2. Comparison of the Intestinal Microbiota Composition

The analysis of species composition is shown in Figure 3 and Figure 4. At the phylum, the predominant bacteria in the hindgut of tilapia included Fusobacteria, Bacteroides, Proteobacteria, Cyanobacteria, Firmicutes. The composition of the predominant bacteria was similar among the groups, but their abundance has varied. Compared with group C, the abundance of Fusobacteria in group H increased, and with the increase in berberine supplementation, the abundance of Fusobacteria first increased and then decreased, with a maximum abundance in the HB2 group (*p* < 0.05). Compared with group C, the Bacteroides in group H increased, and berberine supplementation decreased their abundance; the lowest abundance was in group HB3. Compared with group C, the abundance of Proteobacteria was significantly decreased (*p* < 0.01), and berberine supplementation increased their abundance; the HB3 group had the highest abundance of Proteobacteria, but the abundance of Proteobacteria was still significantly lower than that in group C (*p* < 0.05). The abundance of Firmicutes in the H-HB4 group was significantly lower than that in the C group (*p* < 0.05), and there was no significant difference among the H groups. At the genus level, the dominant bacteria were *Cetobacterium* and *Plesiomonas*, and the abundance of *Cetobacterium* in group H was higher than that in group C. With the increase in berberine addition, the *Cetobacterium* abundance decreased, and the abundance of HB2 was the highest. Compared with group C, the abundance of *Plesiomonas* decreased in group H. Its abundance first increased and then decreased with the increase in berberine content, and the highest abundance was found in group HB2.

#### 3.2.3. Functional Analysis of the Microbial Community in the Intestines

The functional prediction of each group was performed using Tax4fun analysis, and the results are shown in Figure 5. The top 10 functional groups include Membrane Transport, Carbohydrate Metabolism, Amino Acid Metabolism, Energy Metabolism, Metabolism of Cofactors and Vitamins, Signal Transduction, Translation, Nucleotide Metabolism, Replication and Repair, and Lipid Metabolism. At KEGG level 2 (*p* < 0.05), Welch’s *t*-test showed a significant decrease in Carbohydrate Metabolism and Metabolism of Terpenoids and Polyketides, as well as a significant increase in Cell Growth and Death in group H compared to group C (Figure 6a). Membrane Transport and Carbohydrate Metabolism were significantly reduced, and Cell Growth and Death were significantly increased in the HB4 group compared to the C group (Figure 6b). Compared with group H, there was no significant enrichment pathway in group HB4.

### 3.3. Enzyme Activity Related to Bile Acid Synthesis in the Liver

As shown in Figure 7, hepatic bile acid synthase was affected by the high-carbohydrate diet, and the enzyme activities of CYP27A1, CYP8B1 and CYP7A1 were significantly increased in group H compared to group C (*p* < 0.05). All three were also affected by the berberine content, with CYP27A1, CYP8B1 and CYP7A1 decreasing with increasing berberine supplementation.

### 3.4. Bile Acids Content Assay

As shown in Figure 8, the carbohydrate content and berberine supplementation in the diet affected the content of bile acids in the liver and intestines of tilapia, and the content of hepatic and intestinal bile acids increased with berberine supplementation; the content of bile acids in the liver of the HB2-HB4 groups and the content of intestinal bile acid in the HB3-HB4 groups were significantly higher than that of C group (*p* < 0.05).

### 3.5. Serum Biochemical Composition Measurement

As shown in Figure 9, compared with group C, the high-carbohydrate diet significantly increased the content of GLU and TG (*p* < 0.05) in the serum of tilapia, and decreased with the increase in berberine content, with the lowest in HB4 group. T-CHO was not significant.

### 3.6. Liver Morphology

The group C liver cell structure was unaltered, as depicted in Figure 10. The cell’s cytoplasm was plentiful, its form was normal, and its nucleus was situated in the middle. The Group H hepatocytes had enlarged swollen nuclei that were pushed aside by large circular vacuoles. In group HB1, vacuolization became even worse. The vacuoles in the hepatocytes of group HB2-HB4 were reduced in size, the deformation was mitigated, and the arrangement of the cell shapes was relatively normal. Thus indicated that berberine could alleviate lipid deposition in the liver.

### 3.7. Glycogen Content Assay

As shown in Figure 11, an increasing carbohydrate content in the diet caused the accumulation of hepatic glycogen (*p* < 0.05) and muscle glycogen in tilapia; the hepatic glycogen contents further increased with increasing berberine content (*p* < 0.05), with the highest hepatic glycogen content in the HB4 group.

### 3.8. Enzyme Activity Related to Glucose Metabolism in the Liver

The key enzymes of the glucose metabolism were all affected by the carbohydrate content and berberine supplementation. As shown in Figure 12, the activity of HK, PK and PFK increased first and then decreased with the increase in berberine content, and the activity of the HB2 group was the highest (*p* < 0.05). As shown in Figure 13, compared with group C, the activity of PC, PEPCK (*p* < 0.05) and G6Pase was increased in group H. Berberine supplementation decreased the activities of these three gluconeogenic enzymes, and the enzyme activities were the lowest in group HB4.

### 3.9. Gene Expression

The effects of berberine on the glucose metabolism genes of tilapia ingesting high-carbohydrate diets are shown in Figure 14. The expression of HK and PFK increased and then decreased with the increase in berberine content, and the highest expression was in the HB2 group. Compared with group C, the expression of PCK was significantly increased in group H (*p* < 0.05), and the expression of G6PC was also increased by the high-carbohydrate diet; their expression decreased with the increase in the berberine content. GLUT2 expression was up-regulated by the high-carbohydrate diet, and with the increase in the berberine content, GLUT2 firstly increased and then decreased, and the highest expression was found in the HB2 group. Compared with group C, the expression of GYS was up-regulated in group H. Berberine further increased its expression, and group HB4 had the highest expression (*p* < 0.05). In addition, the effect of berberine on bile-acid-related genes in tilapia showed that the expression of CYP7A1 and CYP8B1 decreased with the increase in the berberine content, the expression of FXR increased with the increase in the berberine content, and that the expression of HB4 was significantly higher than that of group C (*p* < 0.05).

### 3.10. Transcriptome Analysis

#### 3.10.1. Sequence Alignment Analysis 

A total of nine samples from three groups, Group C, Group H, and Group HB4, were subjected to transcriptomics. In order to ensure the quality of data, data filtering was performed on the raw data before the information was analyzed in order to reduce the analytical disturbance coming from invalid data. In this experiment, the Q30 value was used as an evaluation standard, and data filtering was conducted on the raw data. The clean data for the nine samples C1, C2, C3, H1, H2, H3, HB4-1, HB4-2, and HB4-3 are 46072128, 35983822, 46672398, 41972638, 46424144, 37946530, 52056750, 44587128, and 45751162, which are 99.01%, 98.77%, 98.16%, 98.63%, 98.62%, 98.63%, 99.23%, 99.15%, and 98.95% of the raw data; all of them are higher than 98%, indicating that the sequencing results can be used for subsequent analysis. The sample relationship was analyzed by taking any two samples and calculating the Pearson correlation coefficient between them, and their correlation is shown in a heat map (Figure 15). The analysis of the heat map shows that all the Pearson coefficients of the samples between the same groups are greater than 0.95, and that the samples in the group are better correlated.

#### 3.10.2. Analysis of Differentially Expressed Genes (DEGs)

The differential gene results for groups C, H, and HB4 are shown in Figure 16. Overall, there were a total of 5297 significantly DEGs, among which the number of significantly differentially expressed increased genes was 2988, accounting for 56.4% of the total number of DEGs; the number of significantly decreased genes was 2309, accounting for 43.6% of the total number of DEGs. C vs. H had 656 up-regulated DEGs and 614 down-regulated DEGs. C vs. HB4 detected the most DEGs, with 1549 up-regulated DEGs and 1161 down-regulated DEGs. H vs. HB4 had 783 up-regulated DEGs and 534 down-regulated DEGs.

#### 3.10.3. GO Enrichment and KEGG Enrichment Analysis of DEGs

The three ontologies of the GO term include molecular function (MF), cellular component (CC), and biological process (BP) (Figure 17). The GO terms significantly enriched in group C, group H, and group HB4 were roughly the same; the biological process was dominated by Cellular process, Single-organism process, and Metabolic process, the molecular function was dominated by Binding, Catalytic activity and Transporter activity, and Cell, Cell part and Organelle dominated the cellular component. Significant enrichment bubble plots (Figure 18) showed that the GO terms with the most significant C vs. H enrichment were Small-molecule metabolic process, Single-organism metabolic process, and Carboxylic acid metabolic process. Metabolic process had the most DEGs, with 775 genes. The most significant enrichment GO terms for C vs. HB4 were Cytoplasm, Cytoplasmic part, Carboxylic acid metabolic process, Cell, and Organic acid metabolic process. Cell had the most DEGs, with 1650 genes. The most significantly enriched GO terms for H vs. HB4 were Cytoplasmic part, Cytoplasm, Intracellular part, Intracellular, etc., and Cell had the most significant gene enrichment, with 691 genes.

The KEGG analysis is shown in Figure 19. There are five KEGG-A levels for Metabolism, Human Diseases, Organismal Systems, Genetic Information Processing, Cellular Processes and Environmental Information Processing. In the KEGG-B level, the Lipid metabolism, there were 64 DEGs in C vs. H, 134 DEGs in C vs. HB4, and 44 DEGs in H vs. HB4. There were 43 DEGs in Carbohydrate metabolism in C vs. H, 95 DEGs in C vs. HB4, and 29 DEGs in H vs. HB4. In Infectious disease: bacterial in human diseases, there were 46 DEGs in C vs. H, 134 DEGs in C vs. HB4, and 68 DEGs in H vs. HB4. In Immune system of Organismal Systems, there were 66 differential genes in C vs. H, 161 DEGs in C vs. HB4, and 84 differential genes in H vs. HB4. Significant enrichment bubble plots (Figure 20) showed that the pathways most significantly enriched for C vs. HB4 were Metabolic pathways, Fatty acid metabolism and others. The highest number of DEGs enriched to Metabolic pathways was 170. The most significantly enriched pathways of C vs. HB4 were Metabolic pathways, Fatty acid degradation, and Carbon metabolism. Among them, the number of DEGs enriched in Metabolic pathways was the highest, with 330. The pathways most significantly enriched for H vs. HB4 were Pyrimidine metabolism, Drug metabolism—other enzymes, Parkinson’s disease, Metabolic pathways, Beta-Alanine metabolism and others. The highest number of DEGs, 125, was enriched for Metabolic pathways.

Through data screening (Table 5), there were 66 DEGs in the glycolysis/gluconeogenesis pathway of C vs. H carbohydrate metabolism, and one DEG in the primary bile acid synthesis pathway of lipid metabolism. There were 20 DEGs in the glycolysis/gluconeogenesis pathway of C vs. HB4. There were three DEGs in the primary bile acid synthesis pathway. There were two DEGs in the H vs. HB4 glycolysis/gluconeogenesis pathway and seven DEGs in the primary bile acid synthesis pathway.

#### 3.10.4. qPCR Verification

To verify the reliability of RNA-Seq sequencing data, 12 DEGs (FABP7, THRSP, RBL, SML, AGXT2, RPL22, CYP2J2, AHSG, HBB, CFHRS, FTL, TDOZA) were randomly selected for qPCR experiments. Six up-regulated and six down-regulated DEGs were included to further validate the gene expression results obtained from the RNA sequencing data. The results showed that the expression patterns of the 12 selected genes were consistent with the RNA sequencing results (Figure 21), further confirming the reliability of RNA-Seq.

### 3.11. Challenge Test

As shown in Figure 22, berberine supplementation in high-carbohydrate diets increased the survival rate of tilapia infected with *Streptococcus agalactiae*, with the highest survival rate in group HB2. Figure 23 showed that SOD, AKP and ACP were significantly reduced in group H compared to group C (*p* < 0.05), while berberine significantly increased the activity of SOD, AKP and ACP, with group HB3 being higher than the other groups.

## 4. Discussion

Carbohydrates, as the cheapest energy source among the three major nutrients, have a protein-saving effect and reduce costs when added to the diet. However, an excessive intake of carbohydrates can cause lipid deposition and liver damage in fish, which will have a negative impact on growth [26]. Berberine is an effective biological component in the traditional Chinese medicine *Coptis chinensis* Franch, which has antibacterial and antipyretic, antioxidant, antiviral, hypoglycemic and hypolipidemic functions [27]. Its synthetic process is mature and inexpensive, and it has been widely used in clinical research. In this experiment, we investigated the effects of berberine on the growth performance, glucose metabolism, bile acid synthesis and intestinal flora of tilapia fed a high-carbohydrate diet. The growth results showed that berberine supplementation in high-carbohydrate diets increased the WGR and SGR, and decreased the HSI and FCR, indicating that berberine supplementation in high-carbohydrate diets in this experiment improved the negative effects caused by a high carbohydrate content and was beneficial to the growth of tilapia.

A high-carbohydrate diet can alter the structure of the intestinal flora, and clinical studies have shown that the ratio of Firmicutes to Bacteroides is positively correlated with blood glucose, and that the ratio of both is significantly raised in patients with diabetes [28]. Similarly, it has been shown that mice raised on a high-carbohydrate diet can experience a significant decrease in their intestinal microbiota, such as Verrucomicrobia and Firmicutes, and an increase in Bacteroides and Actinobacteria [29]. In this experiment, a high-carbohydrate diet decreased the abundance of Firmicutes in the tilapia hindgut and increased the abundance of Bacteroides and Actinobacteria, which was similar to the results of previous studies. Because berberine is not well absorbed orally, it may work its way through the intestinal flora to reduce blood glucose [30]. According to studies, berberine influences the intestinal flora formation [31] and may regulate bile acids through this process, which in turn impacts blood glucose [32]. The action of the intestinal flora on primary bile acids includes deconjugation and dehydroxylation, oxidation or differential isomerization by BSH [33].For example, Firmicutes, which can produce bile salt hydrolase, have the effect of 7α-dehydroxylation [34], and Bacteroidetes, which can produce bile salt hydrolase, can oxidize, epimerize and esterify bile acids [35]. Also demonstrated to enhance glucose metabolism [36] is berberine’s ability to suppress BSH activity [37]. In this experiment, the abundance of bacteria such as Bacteroides decreased after supplementation with berberine, so berberine may regulate the glucose metabolism by inhibiting BSH activity through the inhibition of bacteria such as Bacteroides. The abundance of *Cetobacterium* increased first and then decreased with the increase in berberine supplementation, which may be related to the antibacterial properties of berberine. Functional analysis showed that the ratio of the functional abundance of the carbohydrate metabolism pathway between group C and group H was 0.96, and that the ratio of the functional abundance of the carbohydrate metabolism pathway between group C and group HB4 was 0.82, indicating that berberine improved the disorders related to the glucose metabolism of the intestinal flora of tilapia caused by a high-carbohydrate diet. As previously mentioned, berberine controls the composition of the intestinal flora, particularly by lowering the quantity of flora that produces BSH, which has an impact on the metabolism of carbohydrates in the tilapia’s intestinal flora.

In recent years, the relationship between bile acids and type II diabetes has become a hot research topic in clinical medicine. The sites of bile acid synthesis include the liver and intestinal flora [33]. The liver is the only place where primary bile acids are synthesized, and the intestinal flora converts primary bile acids to secondary bile acids. In this experiment, bile acids were increased in the liver and intestines, suggesting that berberine increased the synthesis of bile acids in the liver and the metabolism of bile acids by the intestinal flora. Hepatic bile acid synthesis is divided into classical and alternative pathways, with CYP7A1 being the rate-limiting enzyme of the classical pathway, which determines three-quarters of bile acid synthesis, and CYP27A1 being the rate-limiting enzyme of the alternative pathway [38]. Bile acids activate FXR, which negatively regulates bile acid synthesis by inducing Small Heterodimer Partner (SHP), thereby inhibiting CYP7A1 expression [39]. CYP7A1 and CYP27A1 are affected by the intestinal flora, and CYP8B1, which determines the proportion of different species of primary bile acids, is not affected by the intestinal flora [14]. It has been shown that the inhibition of CYP8B1 expression promotes 12αhydroxylated bile acid expression, which in turn improves insulin sensitivity [40]. Xiongjie Sun et al. found that berberine restored the disordered intestinal flora, increased bile acids in the intestines, activated the FXR signaling pathway and inhibited the expression of CYP7A1 protein in mice [41]. In this experiment, the high-carbohydrate diet inhibited the expression of FXR and up-regulated the expression of bile-acid-synthesis-limiting enzymes such as CYP7A1. Meanwhile, berberine activates FXR, and the inhibitory effect of FXR on CYP7A1 is enhanced, resulting in an increase in the liver bile acid content. This suggests that berberine improves the insulin resistance of tilapia by activating FXR and increasing the bile acid content. 

Serum glucose is a common indicator of glucose metabolism, and is the main form of glucose transported in organisms; furthermore, triglyceride and the total cholesterol reflect the content of lipids in the blood, and the above indicators are of great significance in the diagnosis of diabetes mellitus and in the treatment of metabolic diseases such as hyperlipidemia. In this experiment, the results of serum biochemical indexes showed that berberine reduced the serum content of GLU, TG and CHO in tilapia reared on high-carbohydrate diets, which was consistent with the findings of Houjun Pan et al. [7]. Glucose is dependent on the glucose transporter (GLUT) for transmembrane transport in most tissue cells, with GLUT2 as an important transporter protein that transports glucose from the cytoplasm to the blood [42]. In this experiment, the berberine increased the expression of GLUT2, improving the efficiency of the conversion of glucose in the blood. An excessive carbohydrate content in the diet can cause lipid deposition in the liver of fish [43]. An increased liver lipid content is manifested as a series of degenerative changes such as the disorganization of hepatocytes, a reduction in the cytoplasm, and the displacement of cell nuclei [44]. Lipid droplets in the liver are eluted into vacuoles by alcohol during the preparation of HE sections. The results of the HE sections of the liver in this experiment showed that the high-carbohydrate feed caused lipid deposition in the liver parenchyma of the tilapia, and that berberine could alleviate liver lipid deposition in tilapia caused by a high carbohydrate content. Glycogen is one of the forms in which organisms store energy. After an organism ingests glycogen, most of it is converted to fat for energy storage, and a small portion is converted to glycogen, which can supply energy quickly compared to fat [45]. Xiaocui Liu et al. showed that berberine maintained the stability of blood glucose by inhibiting the excessive decomposition of hepatic glycogen in diabetic mice and restoring the glycogen structure to its normal form [46]. The results of this experiment on glycogen showed that berberine up-regulated the expression of GYS and increased the content of liver glycogen and muscle glycogen, suggesting that berberine reduces the blood glucose content by promoting the synthesis of liver glycogen and muscle glycogen. Bile acids are one of the targets for the treatment of diabetes, activating FXR to expand the size of the bile acid pool and thereby promoting glycolysis and inhibiting gluconeogenesis [47]. Liver is the key site of glucose metabolism, HK, PK and PFK are the key enzymes of glycolysis, and PEPC, PEPCK and G6Pase are the key enzymes of gluconeogenesis. In this study, glycolytic enzymes and their gene expression increased and then decreased with the rise in berberine content in this experiment, and it is speculated that this may be influenced by the abundance of *Cetobacterium*, which is able to regulate fish glucose homeostasis by acetate [48]. In addition, the inhibition of gluconeogenic enzymes and their gene expression could indicate that berberine lowers blood glucose by inhibiting gluconeogenesis to promote glycolysis. The results of the previous study showed that berberine supplementation in high-carbohydrate diets improved the glucose metabolism of Largemouth bass by promoting glycolysis and inhibiting gluconeogenesis, which is consistent with the results of this experiment [49]. The above results showed that berberine inhibited gluconeogenesis and promoted glycolysis, promoted glycogen synthesis and lowered the blood glucose content, ultimately regulating the glucose metabolism of tilapia.

High-throughput sequencing technology has been widely used in bioscience, such as transcriptomics, metabolomics, immunomics, and intestinal flora analysis, to study the mechanism of growth, metabolism, and immunity of organisms [50]. Among them, transcriptomics is the study of the sequence information and expression information of almost all transcripts of a specific cell, tissue or organ at a specific stage of growth and development or in a certain physiological condition, and it can accurately analyze gene expression differences, structural variations and molecular markers and other issues [51]. For example, Wei Zhang et al. fed diets with different carbohydrate contents to salmon to explore the transcriptome feature of glucose metabolism in the liver [52]. Analysis of the transcriptome data from this experiment revealed that C vs. H and H vs. HB4 differential GOs were enriched most in GO terms such as Cellular process, Single-organism process, and Binding, which are important for the normal life process of the organism, and this result is consistent with other reported results in fish [53]. Analysis of the GO data showed that 55.6% of genes were significantly increased and 44.4% were significantly decreased in C vs. H. In addition, 63.3% of genes were significantly increased and 36.7% were significantly decreased in H vs. HB4. The difference between increased and decreased genes suggests that the carbohydrate content of the diet can have an effect on gene expression in the liver, which may be due to the fact that liver disease induced by a high carbohydrate content in the diet alters the expression of certain genes, and berberine, in turn, alters the expression of certain genes in diseased livers. The expression of the glycolysis/gluconeogenesis pathway in the liver plays a vital role in regulating glucose production and storage and maintaining glucose homeostasis [54]. As one of the signaling molecules that affect glucose homeostasis, the bile acid synthesis pathway is a complex pathway composed of at least 17 enzymes distributed in many organelles. It plays an important role in maintaining cholesterol homeostasis [55]. Xizhan Xu et al. treated diabetic mice with berberine, and the transcriptome results revealed that carbohydrate metabolism, lipid metabolism-related genes and immune response-related genes were significantly enriched in the BBR group, suggesting that BBR treatment could improve lipid metabolism and carbohydrate metabolism, and relieve inflammation in diabetic mice [56]. The results of KEGG pathway enrichment in this experiment showed that the enrichment pathways of the C vs. H group and H vs. HB4 group were mainly concentrated on metabolic pathways, including carbohydrate metabolism and lipid metabolism. A high carbohydrate content increased the number of DEGs in the metabolic pathway, and after berberine supplementation, the DEGs in the metabolic pathway decreased. Compared with C vs. H, the DEGs of the glycolysis/gluconeogenesis pathway in H vs. HB4 were reduced, and the DEGs in the primary bile acid biosynthesis pathway were increased. This may be due to the metabolic disorders caused by the high-carbohydrate diets, which were alleviated by berberine. These results showed that the addition of berberine to the high-carbohydrate diet had a significant effect on carbohydrate metabolism and bile acid synthesis in tilapia. Similarly, the increase in DEGs in the Infectious disease: bacterial and immune system suggests that berberine enhances the immunity of tilapia. 

In recent years, the outbreak of various diseases in aquaculture has caused considerable economic losses to farmers; furthermore the abuse of antibiotics has caused bacterial resistance, drug residues and other problems, so the demand for antibiotic-free aquaculture is increasing. The use of herbs as feed additives can improve the growth and disease resistance of animals [54], and berberine is one of them. *Streptococcus agalactiae* is one of the common pathogenic bacteria in aquaculture, and it can infect Pompano, Grouper, Tilapia and so on [57]. Tilapia infected with *Streptococcus agalactiae* will have a black body, their gill cover margins and pectoral fin bases engorged with blood, and will stray and sping [58]. Superoxide Dismutase (SOD), as an antioxidant enzyme, is able to scavenge harmful free radicals and plays an important role in the immune response of organisms [59]. Acid phosphatase (ACP) and Alkaline phosphatase (AKP) are both important immune enzymes that dephosphorylate nucleic acids [60] and are marker enzymes for macrophage lysosomes that change in response to invasion by pathogens and objects. It was shown that berberine significantly increased SOD activity in insulin resistance [19]; this is consistent with the results of this study. In this experiment, 50 mg/kg of berberine improved the resistance of tilapia infected by *Streptococcus agalactiae*. Hien Van Doan et al. explored the effect of berberine on tilapia infected with *Streptococcus agalactiae* and showed that the addition of 1 g/kg of berberine to the diet increased the survival and disease resistance of tilapia [19]. This was slightly different from our results, probably due to differences in fish weight and diets, but overall, berberine was beneficial to disease resistance in tilapia.

## 5. Conclusions

In summary, this experiment determined the growth performance, serum biochemical indicators, enzymes and genes of glucose metabolism and bile acid, liver transcriptome and intestinal flora of tilapia. It was concluded that berberine affects the glucose metabolism of tilapia by changing the structure of the intestinal flora, stimulating bile acid synthesis, promoting glycolysis and inhibiting gluconeogenesis. Berberine can also improve the disease resistance of tilapia fed high-carbohydrate diets. Therefore, in this experiment, based on the growth rate, the supplementation of 100 mg/kg of berberine in high-carbohydrate diets is beneficial to the growth of tilapia. As a natural product, berberine has few toxic effects and side effects, and has little impact on organisms and the natural environment. More importantly, this study demonstrates that berberine can reduce dietary costs and promote the sustainable development of the aquaculture industry by safely replacing protein with carbohydrates.

## Figures and Tables

**Figure 1 animals-14-01239-f001:**
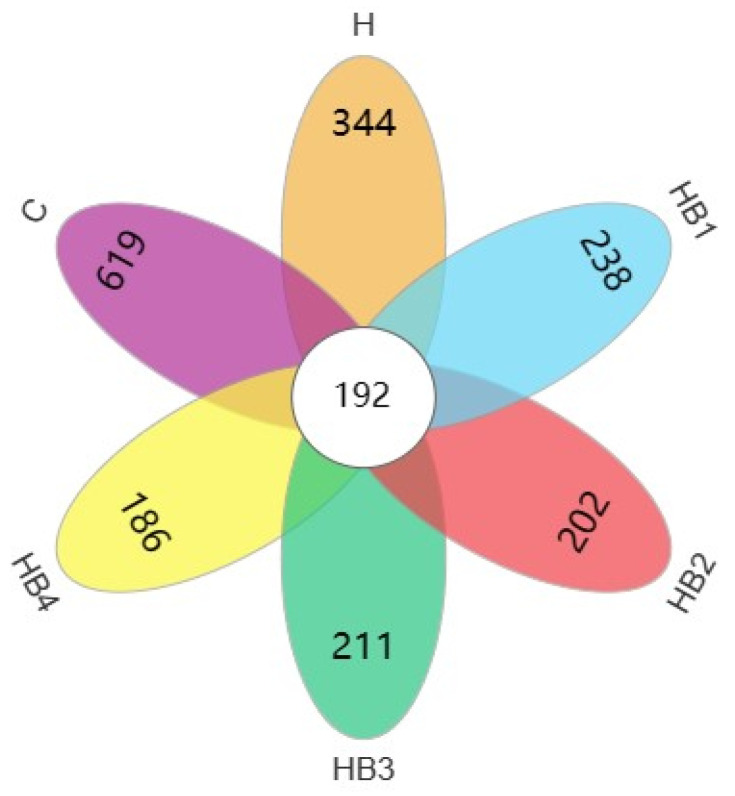
Tilapia hindgut flora species VENN diagram.

**Figure 2 animals-14-01239-f002:**
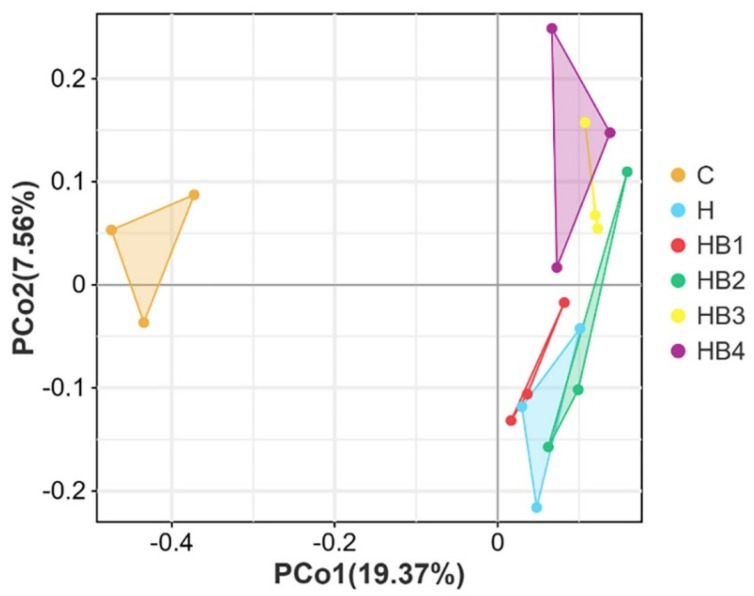
Effects of berberine supplementation in high carbohydrate diets on β diversity of tilapia intestinal flora.

**Figure 3 animals-14-01239-f003:**
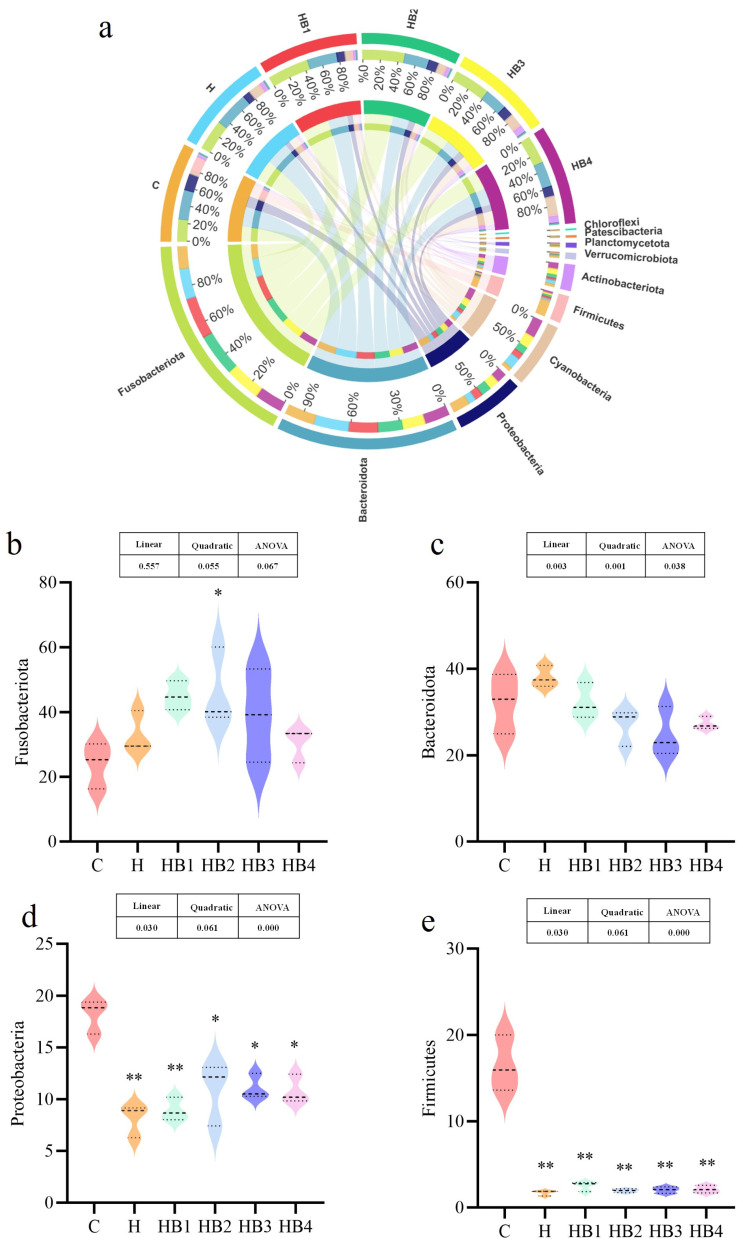
Effects of berberine supplementation in high-carbohydrate diets on tilapia intestinal flora at the phylum level. (**a**) Composition of six groups of phylum-level flora. (**b**) Fusobacteria, (**c**) Bacteroides, (**d**) Proteobacteria, (**e**) Firmicutes. One-way ANOVA and Dunnett’s test analyses (* *p* < 0.05 and ** *p* < 0.01). The three lines in the box plot are the upper quartile, the median and the lower quartile from top to bottom.

**Figure 4 animals-14-01239-f004:**
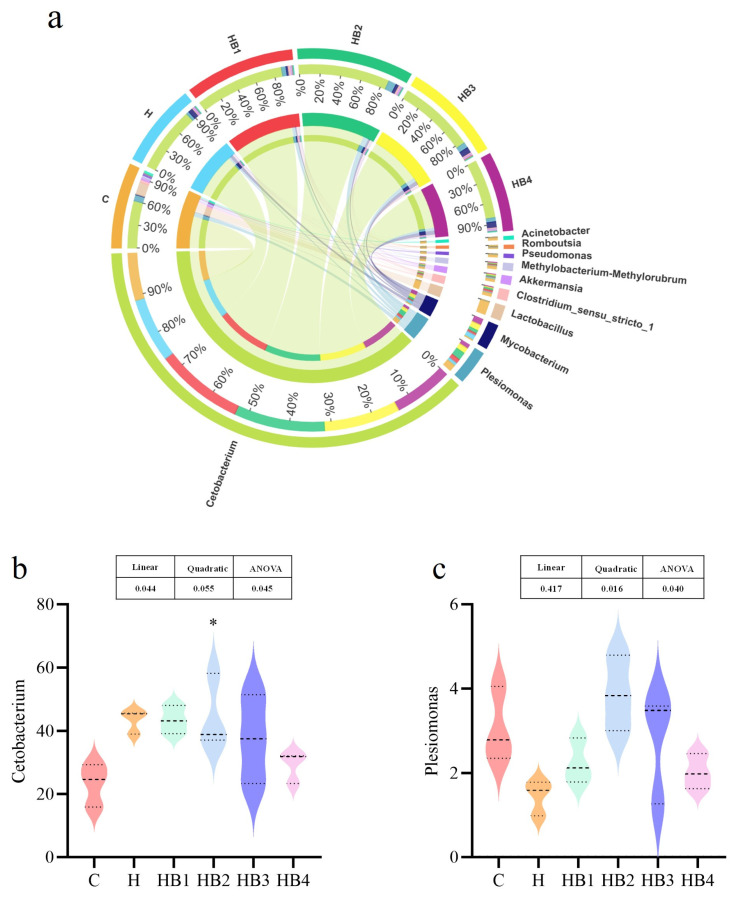
Effects of berberine supplementation in high-carbohydrate diets on tilapia intestinal flora at the genus level. (**a**) Composition of six groups of genus-level flora. (**b**) *Cetobacterium*, (**c**) *Plesiomonas.* One-way ANOVA and Dunnett’s test analyses (* *p* < 0.05). The three lines in the box plot are the upper quartile, the median and the lower quartile from top to bottom.

**Figure 5 animals-14-01239-f005:**
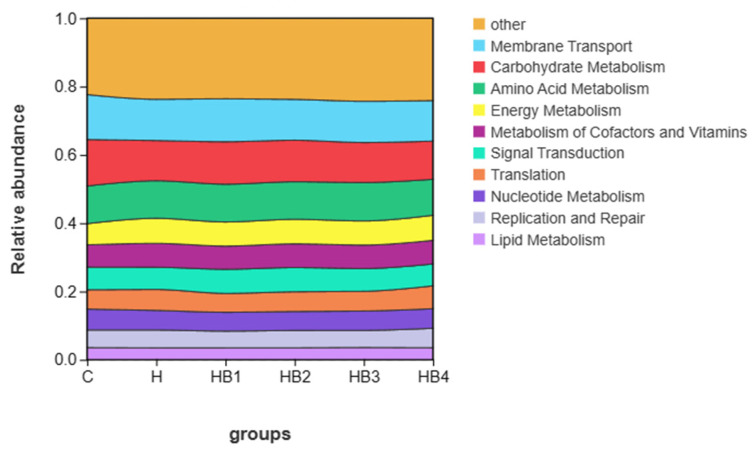
The relative abundance of groups in the intestinal flora of tilapia.

**Figure 6 animals-14-01239-f006:**
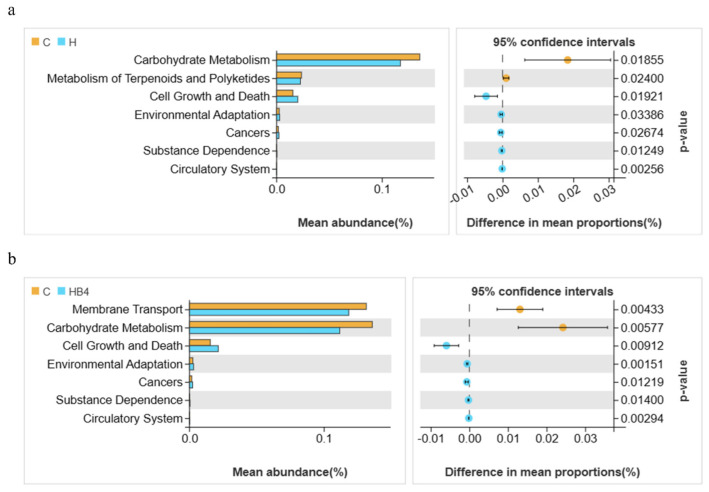
Welch’s T test of the significantly different functions at level 2. (**a**) C vs H; (**b**) C vs HB4.

**Figure 7 animals-14-01239-f007:**
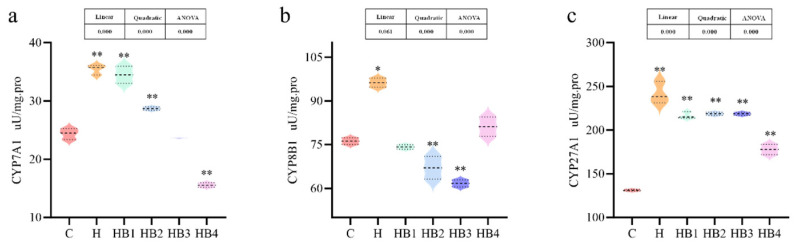
Effect of berberine supplementation in high-carbohydrate diets on the activities of bile acid synthetase in tilapia. (**a**) CYP7A1, cholesterol 7alpha-monooxygenase; (**b**) CYP8B1, sterol-12α-hydroxylase; (**c**) CYP27A1, cholestanetriol 26-monooxygenase. One-way ANOVA and Dunnett’s test analyses (* *p* < 0.05 and ** *p* < 0.01). The three lines in the box plot are the upper quartile, the median and the lower quartile from top to bottom.

**Figure 8 animals-14-01239-f008:**
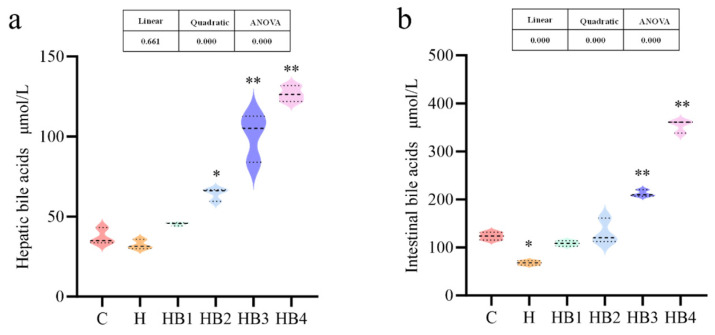
Effects of berberine supplementation in high-carbohydrate diets on liver and intestinal bile acid content in tilapia. (**a**) Hepatic bile acids; (**b**) Intestinal bile acids. One-way ANOVA and Dunnett’s test analyses (* *p* < 0.05 and ** *p* < 0.01). The three lines in the box plot are the upper quartile, the median and the lower quartile from top to bottom.

**Figure 9 animals-14-01239-f009:**
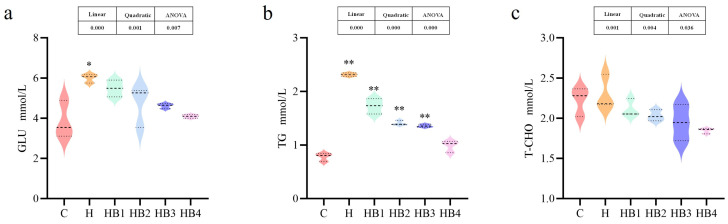
Effect of berberine supplementation in high-carbohydrate diets on serum biochemical indexes of tilapia. (**a**) GLU, glucose; (**b**) TG, triglyceride; (**c**) T-CHO, total cholesterol; One-way ANOVA and Dunnett’s test analyses (* *p* < 0.05 and ** *p* < 0.01). The three lines in the box plot are the upper quartile, the median and the lower quartile from top to bottom.

**Figure 10 animals-14-01239-f010:**
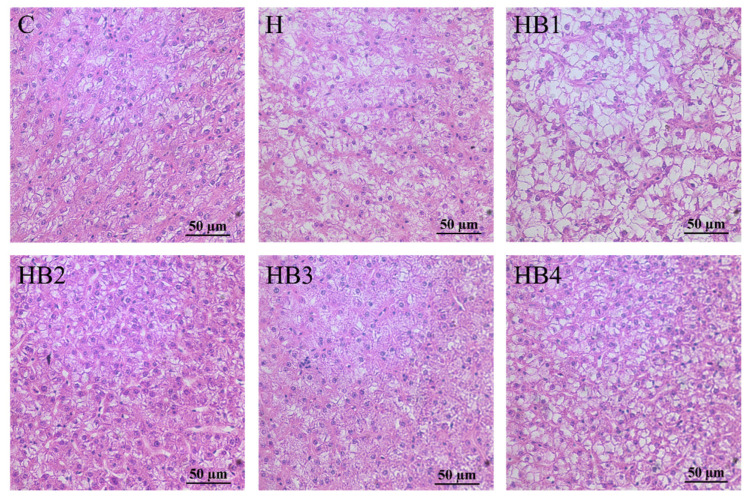
Effect of berberine supplementation in high-carbohydrate diets on liver tissue morphology of tilapia (H.E, ×40 and ×10).

**Figure 11 animals-14-01239-f011:**
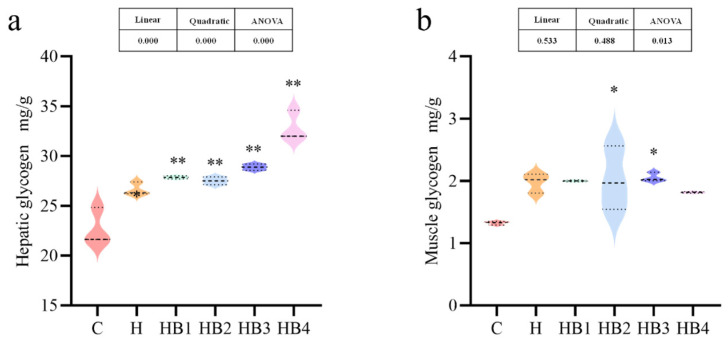
Effects of berberine supplementation in high-carbohydrate diets on hepatic glycogen and muscle glycogen of tilapia. (**a**) Hepatic glycogen; (**b**) muscle glycogen. One-way ANOVA and Dunnett’s test analyses (* *p* < 0.05 and ** *p* < 0.01). The three lines in the box plot are the upper quartile, the median and the lower quartile from top to bottom.

**Figure 12 animals-14-01239-f012:**
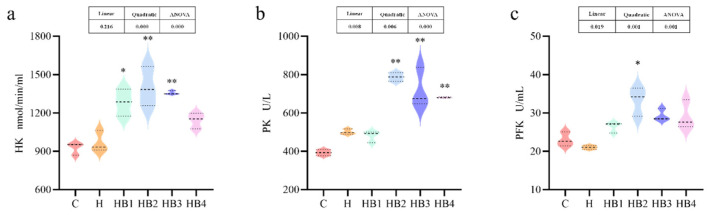
Effect of berberine supplementation in high-carbohydrate diets on the activities of glycolytic enzymes in tilapia. (**a**) HK, hexokinase; (**b**) PK, pyruvate kinase; (**c**) PFK, phosphofructokinase. One-way ANOVA and Dunnett’s test analyses (* *p* < 0.05 and ** *p* < 0.01). The three lines in the box plot are the upper quartile, the median and the lower quartile from top to bottom.

**Figure 13 animals-14-01239-f013:**
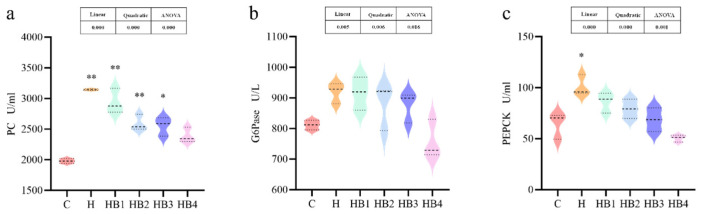
Effect of berberine supplementation in high-carbohydrate diets on the activities of gluconeogenic enzymes in tilapia. (**a**) PC, pyruvate carboxylase; (**b**) G6Pase, glucose-6-phosphatase; (**c**) PEPCK, phosphoenolpyruvate carboxykinase. One-way ANOVA and Dunnett’s test analyses (* *p* < 0.05 and ** *p* < 0.01). The three lines in the box plot are the upper quartile, the median and the lower quartile from top to bottom.

**Figure 14 animals-14-01239-f014:**
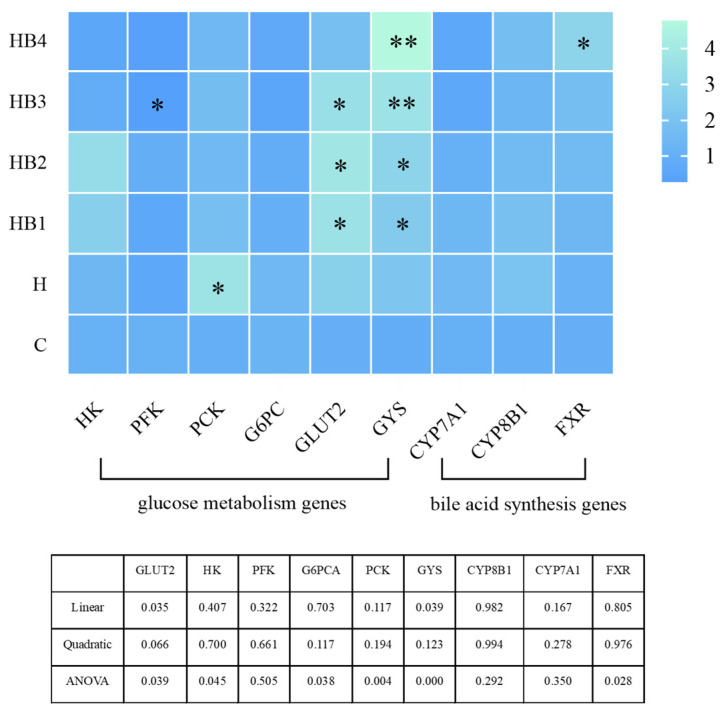
Effects of berberine supplementation in high-carbohydrate diets on glucose metabolism genes and bile acid synthesis genes in tilapia. HK, hexokinase; PFKl, phosphofructokinase, liver-type gene; G6PC, glucose-6-phosphatase; PCK, phosphoenolpyruvate carboxykinase; GYS, glycogen synthase; GLUT2, protein O-glucosyltransferase 2; CYP7A1, cholesterol 7alpha-monooxygenase; CYP8B1, sterol-12α-hydroxylase; FXR, farnesoid X receptor. One-way ANOVA and Dunnett’s test analyses (* *p* < 0.05 and ** *p* < 0.01).

**Figure 15 animals-14-01239-f015:**
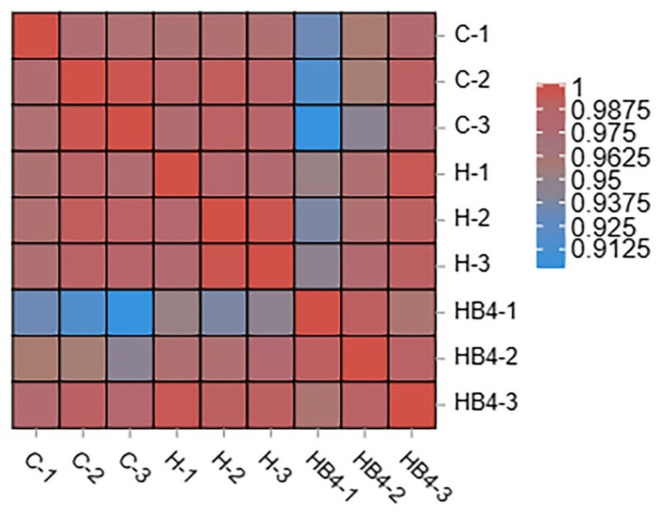
Correlation heatmap of liver samples from group C, group H and group HB4.

**Figure 16 animals-14-01239-f016:**
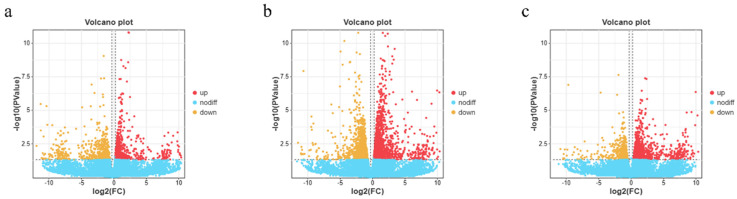
Effect of berberine supplementation in high-carbohydrate diets on the volcano map of DEGs of tilapia in (**a**) C vs. H, (**b**) C vs. HB4, and (**c**) H vs. HB4.

**Figure 17 animals-14-01239-f017:**
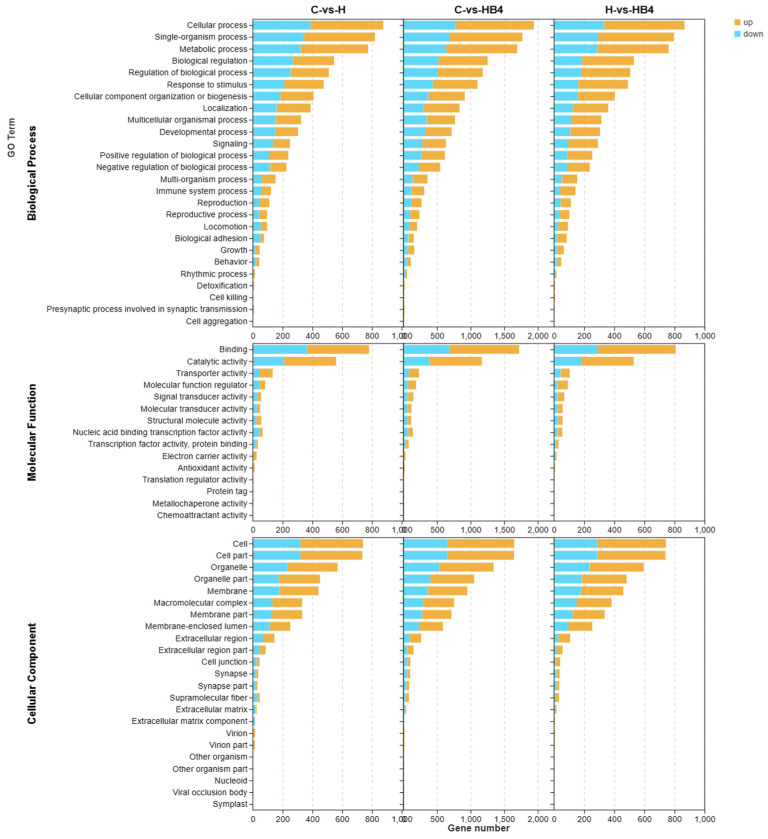
GO term functional enrichment analysis of DEGs in C vs. H, C vs. HB4, and H vs. HB4.

**Figure 18 animals-14-01239-f018:**
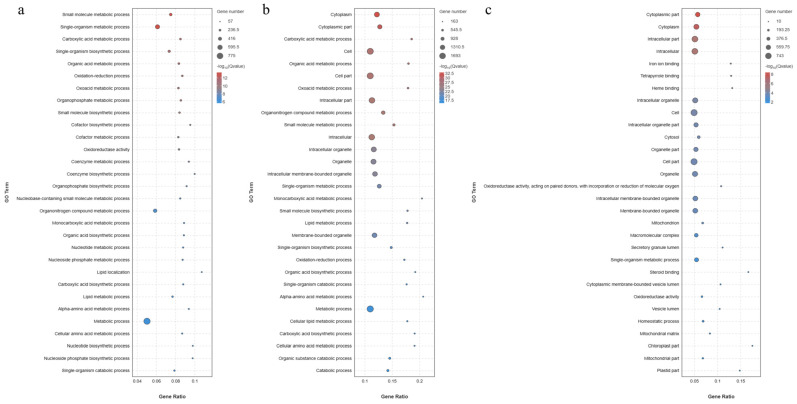
GO term functional enrichment analysis of DEGs in (**a**) C vs. H, (**b**) C vs. HB4, and (**c**) H vs. HB4.

**Figure 19 animals-14-01239-f019:**
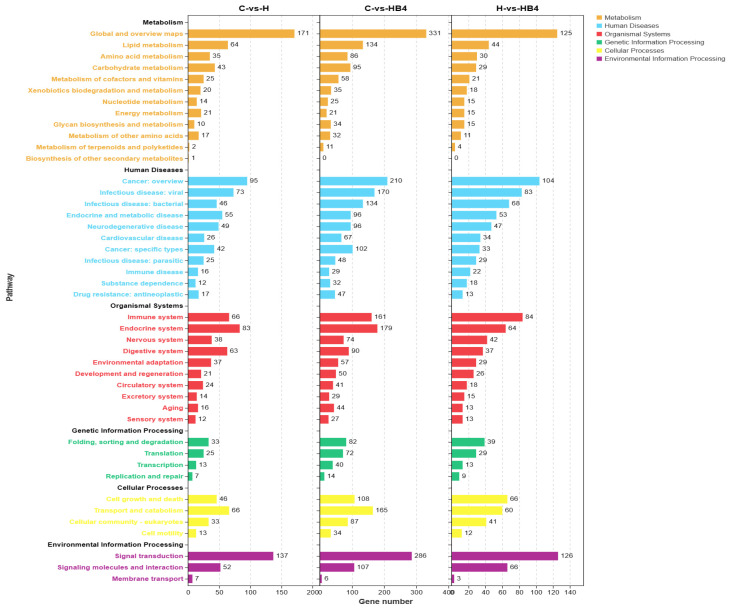
KEGG functional enrichment analysis of DEGs in C vs. H, C vs. HB4, and H vs. HB4.

**Figure 20 animals-14-01239-f020:**
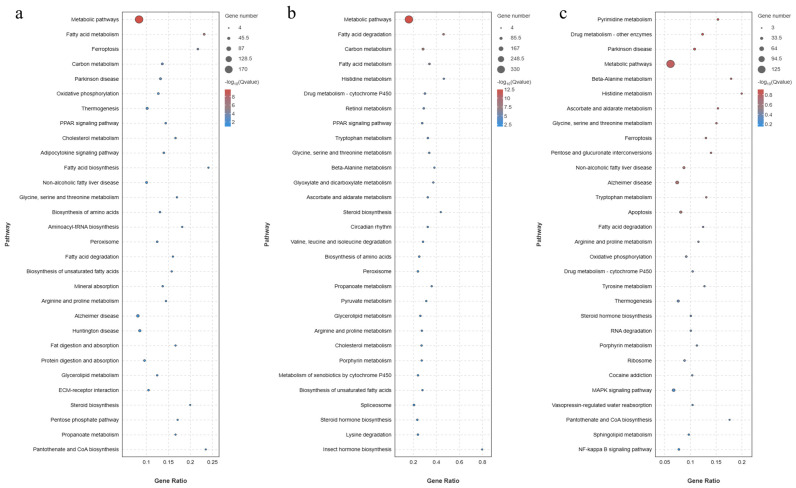
KEGG functional enrichment analysis of DEGs in (**a**) C vs. H, (**b**) C vs. HB4, and (**c**) H vs. HB4.

**Figure 21 animals-14-01239-f021:**
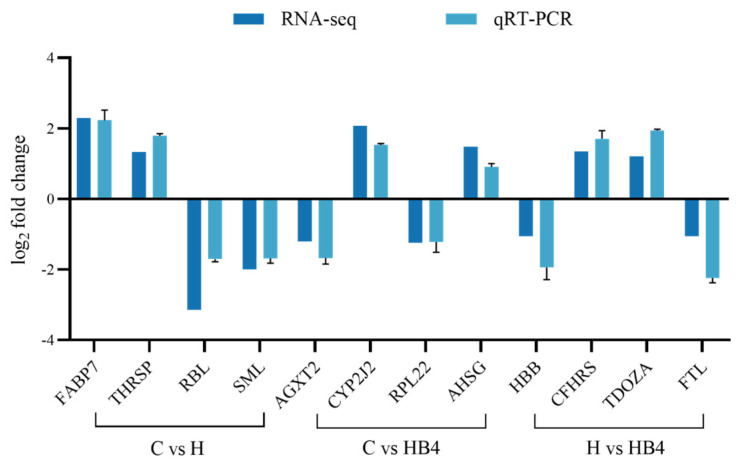
Validation of DEGs by qPCR.

**Figure 22 animals-14-01239-f022:**
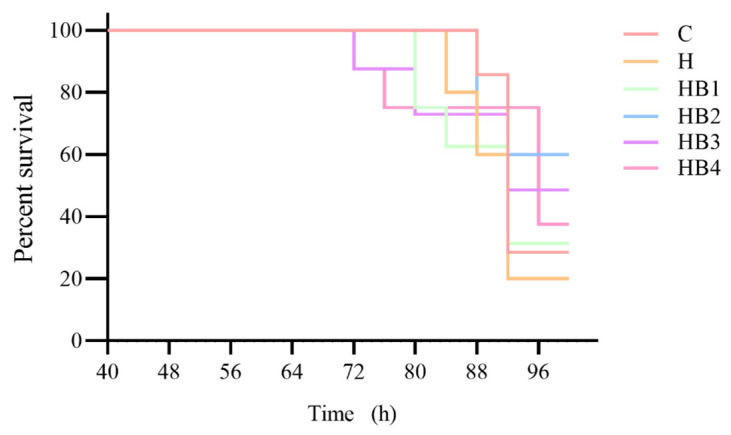
*Streptococcus agalactiae* on percent survival in the challenge test of tilapia.

**Figure 23 animals-14-01239-f023:**
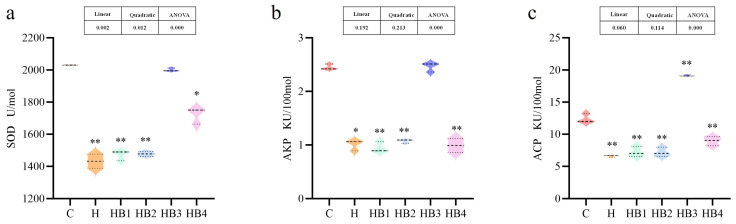
Effect of berberine supplementation in high-carbohydrate diets on the activities of SOD, AKP and ACP in the serum of tilapia after the injection of *Streptococcus agalactiae.* (**a**) SOD, superoxide dismutase; (**b**) AKP, alkaline phosphatase; (**c**) ACP, acid phosphatase. One-way ANOVA and Dunnett’s test analyses (* *p* < 0.05 and ** *p* < 0.01). The three lines in the box plot are the upper quartile, the median and the lower quartile from top to bottom.

**Table 1 animals-14-01239-t001:** Formulation and proximate analysis of trial diets (dry matter, DM %).

Ingredients	C	H	H1	H2	H3	H4
white fishmeal	25	25	25	25	25	25
casein	16	16	16	16	16	16
gelatin	4	4	4	4	4	4
corn starch	25	40	40	40	40	40
paste starch	4	4	4	4	4	4
fish oil	5	5	5	5	5	5
soybean lecithin	1	1	1	1	1	1
vitamin C	0.1	0.1	0.1	0.1	0.1	0.1
Premix ^a^	1	1	1	1	1	1
Ca(H_2_PO_4_)_2_	2	2	2	2	2	2
choline chloride	0.5	0.5	0.5	0.5	0.5	0.5
ethoxyquin	0.03	0.03	0.03	0.03	0.03	0.03
attractant	0.1	0.1	0.1	0.1	0.1	0.1
berberine	0	0	0.0025	0.005	0.0075	0.01
microcrystalline cellulose	16.27	1.27	1.2675	1.265	1.2625	1.26
total	100	100	100	100	100	100
crude protein	40.91	41.67	40.89	40.61	40.63	41.53
crude lipid	6.94	6.14	6.25	6.61	5.89	5.66
ash	7.33	7.27	7.34	7.4	7.33	7.34
carbohydrate	27.38	41.05	40.96	41.39	40.31	40.91

^a^. Each 1 kg of premix contains the following: vitamins A 300,000 IU, vitamins B2 0.45 g, vitamins B6 0.15 g, vitamins B12 0.0009 g, vitamins C 50.0 g, vitamins D3 35,000 IU, vitamins K3 0.08 g, niacinamide 2.0 g, biotin 0.004 g, calcium pantothenate (CH_2_O_4_) 0.9 g, CuSO_4_ 3.0 g, FeSO_4_ 30.0 g, ZnSO_4_ 20.0 g, MnSO_4_ 2.4 g.

**Table 2 animals-14-01239-t002:** Primers used in the real-time PCR analysis.

Gene	Sequence 5′-3′	Reference/Accession No.
HK-F	CAGCACGGAACTCCATGATGACC	(Yong-Jun Chen, 2017) [25]
HK-R	GCACAAATGTGGGCAGCATCTTG	(Yong-Jun Chen, 2017) [25]
PFKl-F	GCCGCCTTCAACCTGGTTAAGA	(Yong-Jun Chen, 2017) [25]
PFKl-R	GTGCCGCAGAAGTCGTTGTCTA	(Yong-Jun Chen, 2017) [25]
G6PC-F	ATCGGAGACTGGCTCAACTTGGT	(Yong-Jun Chen, 2017) [25]
G6PC-R	TGGCATGACCTGAAGGACTTCCT	(Yong-Jun Chen, 2017) [25]
PCK-F	CGAGTGGAGAGCAAGACTGTGA	(Yong-Jun Chen, 2017) [25]
PCK-R	TTGGTGAGAGCTGAGCCTACG	(Yong-Jun Chen, 2017) [25]
GYS-F	TGACAAGGAGGCTGGTGAGAGG	(Yong-Jun Chen, 2017) [25]
GYS-R	ACTCGTGCATGGCTGAGAACTT	(Yong-Jun Chen, 2017) [25]
GLUT2-F	GGCACTCTAGCTCTGGCTGTGT	(Yong-Jun Chen, 2017) [25]
GLUT2-R	GGGTGGTGACCTGGGTCTTCTT	(Yong-Jun Chen, 2017) [25]
CYP7A1-F	GGGATAAGACACAGGCAACCA	XM_003456729.5
CYP7A1-R	TGCGGAGGAATTGAAGTGGG	XM_003456729.5
CYP8B1-F	GGGTAAAGAGGAGATGGGAATG	XM_019348226.2
CYP8B1-R	GGAGCAGCCAGAATGAAGAA	XM_019348226.2
FXR-F	AACCATCCTGACACCAGATCG	XM_025899550.1
FXR-R	CAACAGAGGCTGGGAAAGGA	XM_025899550.1
βactin	CAGTGCCCATCTACGAG	(Yong-Jun Chen, 2017) [25]
βactin	CCATCTCCTGCTCGAAGTC	(Yong-Jun Chen, 2017) [25]

HK, hexokinase; PFKl, phosphofructokinase, liver-type gene; G6PCA, glucose-6-phosphatase; PCK, phosphoenolpyruvate carboxykinase; GYS, glycogen synthase; GLUT2, protein O-glucosyltransferase 2; CYP7A1, cholesterol-7α-monooxygenase; CYP8B1, sterol-12α-hydroxylase; FXR, farnesoid X receptor.

**Table 3 animals-14-01239-t003:** Effect of berberine supplementation in high-carbohydrate diets on the growth performance of tilapia.

Group	C	H	HB1	HB2	HB3	HB4	Linear	Quadratic	ANOVA
SR (%)	98.90 ± 1.91	97.80 ± 1.91	97.80 ± 1.91	95.56 ± 1.96	94.46 ± 6.93	93.33 ± 6.65	0.118	0.307	0.566
WGR (%)	1486.08 ± 16.57	1433.87 ± 35.70	1435.85 ± 30.36	1448.69 ± 5.36	1495.85 ± 17.22	1501.53 ± 8.65	0.001	0.004	0.021
SGR (%)	5.03 ± 0.02	4.96 ± 0.06	4.97 ± 0.04	4.98 ± 0.01	5.04 ± 0.02	5.37 ± 0.34	0.075	0.095	0.332
FCR	0.97 ± 0.01	1.00 ± 0.03	1.00 ± 0.02	0.99 ± 0.00	0.96 ± 0.01	0.96 ± 0.00	0.002	0.008	0.031
VSI (%)	13.38 ± 0.67	11.55 ± 0.57	14.18 ± 1.50	14.35 ± 0.80	14.10 ± 0.48	13.87 ± 0.45	0.135	0.060	0.229
HIS (%)	1.56 ± 0.07	1.79 ± 0.20	2.21 ± 0.15 *	2.12 ± 0.09 *	1.74 ± 0.06	1.36 ± 0.10	0.042	0.001	0.003
CF (g/cm^3^)	3.67 ± 0.05	3.96 ± 0.05	3.57 ± 0.15	3.63 ± 0.08	3.83 ± 0.24	3.72 ± 0.05	0.645	0.399	0.305

SR, Survival rate; WGR, Weight gain rate; SGR, Specific growth rate; FCR, Feed conversion ratio; HSI, Hepatosomatic ratio; VSI, Viscerosomatic index; CF, Condition factor. One-way ANOVA and Dunnett’s test analyses (* *p* < 0.05).

**Table 4 animals-14-01239-t004:** Effects of berberine supplementation in high-carbohydrate diets on α diversity of tilapia intestinal flora.

Group	C	H	HB1	HB2	HB3	HB4	Linear	Quadratic	ANOVA
sobs	788.67 ± 107.29	544.67 ± 2.52	511.00 ± 70.15 *	447.67 ± 28.59 **	483.00 ± 67.01 *	490.33 ± 93.29 *	0.238	0.198	0.001
Chao 1	854.87 ± 79.71	611.64 ± 17.51 *	619.00 ± 65.14 **	514.86 ± 15.39 **	568.23 ± 55.88 **	564.34 ± 74.53 **	0.275	0.186	0.000
ace	903.58 ± 57.91	592.33 ± 5.66 **	612.82 ± 46.76 **	515.50 ± 11.15 **	558.20 ± 66.89 **	571.58 ± 60.36 **	0.553	0.199	0.000
Shannon	4.97 ± 1.23	3.63 ± 0.63	3.21 ± 0.66	3.22 ± 0.85	3.61 ± 1.19	4.23 ± 0.44	0.183	0.238	0.195
Simpson	0.86 ± 0.08	0.80 ± 0.06	0.74 ± 0.07	0.71 ± 0.14	0.75 ± 0.18	0.86 ± 0.05	0.325	0.338	0.435

One-way ANOVA and Dunnett’s test analyses (* *p* < 0.05 and ** *p* < 0.01).

**Table 5 animals-14-01239-t005:** DEGs enriched in metabolic pathways.

Genes	Pathway	Gene ID	Log2 (fc)
**C-vs-H**			
Pgm1	Glycolysis/Gluconeogenesis	ncbi_100534546	0.29
ENO3	Glycolysis/Gluconeogenesis	ncbi_100699677	0.46
ENO1	Glycolysis/Gluconeogenesis	ncbi_100701182	2.34
ADH5	Glycolysis/Gluconeogenesis	ncbi_100702001	0.51
GPI	Glycolysis/Gluconeogenesis	ncbi_100704840	4.32
G6PC	Glycolysis/Gluconeogenesis	ncbi_100709522	0.46
Cyp46a1	Primary bile acid biosynthesis	ncbi_100703511	7.92
**C-vs-HB4**			
tpi1b	Glycolysis/Gluconeogenesis	ncbi_100534540	0.52
Pgm1	Glycolysis/Gluconeogenesis	ncbi_100534546	0.43
PDHA1	Glycolysis/Gluconeogenesis	ncbi_100691217	−1.71
Acss2	Glycolysis/Gluconeogenesis	ncbi_100692076	2.60
Aldh3a2	Glycolysis/Gluconeogenesis	ncbi_100693723	0.57
ADH5	Glycolysis/Gluconeogenesis	ncbi_100695823	−0.86
ACSS2	Glycolysis/Gluconeogenesis	ncbi_100697261	0.74
Aldh3a2	Glycolysis/Gluconeogenesis	ncbi_100698670	0.59
ENO3	Glycolysis/Gluconeogenesis	ncbi_100699677	0.47
Pgk1	Glycolysis/Gluconeogenesis	ncbi_100701118	0.74
ENO1	Glycolysis/Gluconeogenesis	ncbi_100701182	2.79
ADH5	Glycolysis/Gluconeogenesis	ncbi_100702001	0.57
ADH5	Glycolysis/Gluconeogenesis	ncbi_100702534	−2.03
ADH5	Glycolysis/Gluconeogenesis	ncbi_100702802	0.34
PFKM	Glycolysis/Gluconeogenesis	ncbi_100702997	0.38
ALDH16A1	Glycolysis/Gluconeogenesis	ncbi_100705854	1.08
ALDH2	Glycolysis/Gluconeogenesis	ncbi_100707184	0.51
G6PC	Glycolysis/Gluconeogenesis	ncbi_100709522	0.76
aldh9A1	Glycolysis/Gluconeogenesis	ncbi_109194280	0.34
aldh9A1	Glycolysis/Gluconeogenesis	ncbi_109196929	0.47
HSD3B7	Primary bile acid biosynthesis	ncbi_100534450	0.52
Cyp46a1	Primary bile acid biosynthesis	ncbi_100692681	0.55
CYP8B1	Primary bile acid biosynthesis	ncbi_109195682	0.78
**H-vs-HB4**			
ALDH16A1	Glycolysis/Gluconeogenesis	ncbi_100705854	0.75
aldh9A1	Glycolysis/Gluconeogenesis	ncbi_109196929	0.51
HSD3B7	Primary bile acid biosynthesis	ncbi_100534450	0.79
CYP8B1	Primary bile acid biosynthesis	ncbi_100693750	0.49
CYP7B1	Primary bile acid biosynthesis	ncbi_100694909	1.48
Hsd17b4	Primary bile acid biosynthesis	ncbi_100706762	0.36
AKR1D1	Primary bile acid biosynthesis	ncbi_109194203	0.18
ch25h	Primary bile acid biosynthesis	ncbi_109194213	−0.41
CYP8B1	Primary bile acid biosynthesis	ncbi_109195682	0.78

## Data Availability

The data presented in this study are available in the main article.

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
