# Peer review of "The Supplementation of Berberine in High-Carbohydrate Diets Improves Glucose Metabolism of Tilapia (Oreochromis niloticus) via Transcriptome, Bile Acid Synthesis Gene Expression and Intestinal Flora"

_animals, 2024, doi:10.3390/ani14081239_

Round 1
Reviewer 1 Report
Comments and Suggestions for Authors
The article is devoted to the use of biologically active additive - berberine in feeding tilapia at high content of carbohydrates in compound feed. This research has high scientific and practical significance and contributes to the development of world aquaculture and feed production.
Despite the high degree of preparation of the article, there are minor remarks:
The article discusses feeds with high carbohydrate content: what amount are we talking about? At what carbohydrate content is it recommended to add berberine in the amount of 100 mg/kg? Table 1 shows the main feed parameters - moisture, crude protein, crude lipid - but no carbohydrate content.
Why was the range from 0 to 0.01% of berberine in the feed chosen? Is there a possibility that increasing the berberine content to a greater extent will increase the values obtained? This should be reflected in the paper.
The Materials and Methods describes the methodology for determining survival rate, growth rate, feed conversion ratio and other (lines 127-134), but these are not reflected in the results and conclusions.
After elimination of the above remarks, the article can be published in Animals.
Reviewer 2 Report
Comments and Suggestions for Authors
Most tests performed used commercial analytical kits. I suggest adding a note or a sentence indicating the reliability of these commercial tests compared to traditional laboratory procedures for the analysis. A citation of a paper that used similar kits can also indicate acceptance of the use of commercial kits in this type of research.
The English is readable and understandable, but it has some flaws regarding flow, accuracy, and clarity. It needs some review to improve readability. Some specific examples regarding English editing and other recommendations are below:
Line 18 - Incomplete sentence. It’s missing a verb.
Line 57 – Punctuation adjustment.
Line 104 – The sentence can be improved. The word mix is presented 4 times in different forms.
Line 110 - 18 1 m3 is visually confusing to read.
Table 1— More clarification is needed on why the microcrystalline cellulose content is 10x higher in normal formulation (group C) than in high-carb formulation.
Line 220-222 – The sentence is very confusing.
Line 265-266 – Similar to the previous example. When did the increase and decrease happen? An indication of time would help to clarify the sentence.
Line 274 – The structure of this sentence is poor. Suggestion: With the addition of berberine, the abundance of (what?) decreased and ….
Figures 3, 4, 6, 8, 11. The caption should identify parts a, b, c, d, and e.
Line 500 – you can delete the second carbohydrates
Line 526. Punctuation.
Lines 541-546 – Rewrite to improve flow, punctuation, and avoid repetition.
Line 555 – sensitivity.
Lines 558-563 – This sentence is too long.
Line 576 – replace , for . after citation.
Line 667 – repetition of “In this experiment”
The conclusion can be improved, especially in this part: “So supplementing 100mg/kg 680 berberine in high carbohydrate diets is beneficial to the growth of tilapia.” I suggest adding more elements explaining why you are recommending the highest dosage tested, such as the performance of the HB4 group, or even adding economic insight, as it was mentioned that berberine is inexpensive. I was expecting to read in the conclusion that berberine can reduce the cost of diets by safely replacing protein content with carbohydrates or something around this idea. Line 511 presents a strong concluding statement that can be used. Line 677, the word “may” should be deleted to avoid imprecision.
Overall, this paper offers high-quality research that is highly applicable to the aquaculture industry. It contributes to enhancing productivity and health and reducing production costs of tilapia. I recommend accepting this manuscript after minor adjustments.
Comments on the Quality of English LanguageThe English is readable and understandable, but it has some flaws regarding flow, accuracy, and clarity. It needs some review to improve readability. Specific recommendations regarding English editing were provided in the previous section of general comments to the authors.
Reviewer 3 Report
Comments and Suggestions for Authors
The manuscript entitled “The supplement of berberine in high carbohydrate diets improve glucose metabolism of tilapia (Oreochromis niloticus) via transcriptome, bile acid synthesis DEGs and intestinal bacteria” by Hongyu Liu et al. reports the data on the effects of berberine (a plant-derived alkaloid) supplementation on the transcriptome, microbiome and physiology of tilapia fed a high carbohydrate diet. From a fish production perspective, the study contributes to solving the problem of formulating growth-stimulating diets, particularly carbohydrate-enriched.
Basic reporting
In general, the experiment is well designed and provides new data. The overall panel of parameters used to describe the effects of berberine on blood biochemistry and hepatic metabolic activity, in particular glucose metabolism and tolerance in tilapia, seems adequate and relevant to the study. As far as I can evaluate, the sampling and experimental procedures are well designed and quite detailed. The preliminary hypothesis was justified by the data summarised in 23 figures and five tables; the presentation is of high quality, clear and graphically well organised. The main finding is that berberine supplementation to a diet enriched with carbohydrates (starch and maize starch) can affect the glucose metabolism in tilapia shifting the intestinal flora composition, stimulating bile acid synthesis, accelerating glycolysis and inhibiting gluconeogenesis. The most beneficial dose was found to be the maximal of the studied, 100 mg per kg of feed. The growth stimulating effect of a high carbohydrate diet was also confirmed. The conclusion on the effect of berberine in improving the disease resistance in tilapia fed a high carbohydrate diet is supported be the data provided on the S. agalactiae infection challenge test. The statistical analyses applied are appropriate. The text of the MS is clearly written. In general, the MS is suitable for publication in ‘Animals’, although I have a few minor comments detailed below.
Minor comments
Line 4, misprint in the title: DGEs instead of DEGs
Lines 17-19, missing verb, please check
Line 62, ‘material and energy metabolism’ better replace with ‘plastic and energy metabolism’
Lines 85-86, please check for punctuation and consistency (here, better ‘belonging’ instead of ‘belongs’, etc.)
Line 88, a supporting reference to the carbohydrate requirement of tilapia is desirable.
Line 101, please specify berberine formulae (perhaps disulphate? hydrochloride?); according to the description “berberine (HPLC ≥ 99%, Maclean's)”, the worst bioavailable form of berberine was used. Is it so?
Line 102, please provide justification for the chosen dosages of berberine used in the study.
Lines 526-529, please check the phrase for punctuation and consistency (verbs after ‘BSH’).
Summarising the strengths and areas for improvement of the MS, I recommend a minor revision according to the above recommendations.
Author Response
请参阅附件
